# Prevalence of Gastrointestinal Parasites in Zoo Animals and Phylogenetic Characterization of *Toxascaris leonina* (Linstow, 1902) and *Baylisascaris transfuga* (Rudolphi, 1819) in Jiangsu Province, Eastern China

**DOI:** 10.3390/ani14030375

**Published:** 2024-01-24

**Authors:** Weimin Cai, Yu Zhu, Feiyan Wang, Qianqian Feng, Zhizhi Zhang, Nianyu Xue, Xun Xu, Zhaofeng Hou, Dandan Liu, Jinjun Xu, Jianping Tao

**Affiliations:** 1College of Veterinary Medicine, Yangzhou University, Yangzhou 225009, China; wmcaii@126.com (W.C.); zy18362822874@163.com (Y.Z.); dx120210181@stu.yzu.edu.cn (F.W.); qianqianfeng98@163.com (Q.F.); tiffany_zzz@163.com (Z.Z.); yznianyuxue@126.com (N.X.); zfhou@yzu.edu.cn (Z.H.); ddliu@yzu.edu.cn (D.L.); jjxu@yzu.edu.cn (J.X.); 2Jiangsu Co-Innovation Center for Prevention and Control of Important Animal Infectious Diseases and Zoonoses, Yangzhou University, Yangzhou 225009, China; 3Jiangsu Key Laboratory of Zoonosis, Yangzhou University, Yangzhou 225009, China; 4Yangzhou Zhuyuwan Zoo, Yangzhou 225009, China; yzboyxx@163.com

**Keywords:** wildlife, parasite, investigation, *Toxascaris leonina*, *Baylisascaris transfuga*, phylogenetics

## Abstract

**Simple Summary:**

A study in eastern China has found that over 40% of zoo animals are infected with gastrointestinal parasites, which pose a threat to their welfare and the health of visitors and veterinarians. More than 11 parasite species were identified in the study conducted at Zhuyuwan Zoo, including two species of *Ascaris*, and they detected *Paramphistomum* spp. eggs in the captive Père David’s deer and *Fasciola* spp. eggs in sika deer, which had not been previously reported in Chinese zoos. The study highlights the need for prevention and control measures to be implemented to tackle parasitic diseases in zoo animals.

**Abstract:**

The burden of gastrointestinal parasites in zoo animals has serious implications for their welfare and the health of veterinarians and visitors. Zhuyuwan Zoo is located in the eastern suburb of Yangzhou city in eastern China, in which over 40 species of zoo animals are kept. In order to understand the infection status of GI parasites in Zhuyuwan Zoo, a total of 104 fresh fecal samples collected randomly from birds (*n* = 19), primates (*n* = 19), and non-primate mammals (*n* = 66) were analyzed using the saturated saline flotation technique and nylon sifter elutriation and sieving method for eggs/oocysts, respectively. Two *Ascaris* species were molecularly characterized. The results showed that the overall prevalence of parasitic infection was 42.3% (44/104). The parasitic infection rate in birds, primates, and non-primate mammals were 26.3% (5/19), 31.6% (6/19), and 50.0% (33/66), respectively. A total of 11 species of parasites were identified, namely, Trichostrongylidae, *Capillaria* sp., *Trichuris* spp., *Strongyloides* spp., *Amidostomum* sp., *Toxascaris leonina*, *Baylisascaris transfuga*, *Parascaris equorum*, *Paramphistomum* spp., *Fasciola* spp., and *Eimeria* spp. *Paramphistomum* spp. eggs were first detected from the captive Père David’s deer, and *Fasciola* spp. eggs were first reported from sika deer in zoo in China. A sequence analysis of ITS-2 and *cox*1 showed that the eggs isolated from the African lion (*Panthera leo* Linnaeus, 1758) were *T. leonina*, and the eggs from the brown bear (*Ursus arctos* Linnaeus, 1758) were *B. transfuga*. The public health threat posed by these potential zoonotic parasitic agents requires attention. These results lay a theoretical foundation for prevention and control of wild animal parasitic diseases at zoos in China.

## 1. Introduction

Wild animals are not only important assets to the natural world but are also considered companions to humans. Nonetheless, the rapid increase in the global human population over the past six decades has resulted in significant ecological changes and a loss of wildlife habitats [1]. Consequently, the survival of wild animals has been jeopardized, with certain species teetering on the brink of extinction [2,3]. Additionally, both captive and wild animals contribute to the spread of various diseases. Keeping wild animals in zoos can worsen the problem of parasitic infections, posing a significant threat to endangered species and sometimes causing unexpected declines in local populations [4].

The prevalence of GI parasites in zoo animals poses a significant health concern, with symptoms such as apathy, colic, diarrhea, malaise, and weight loss [5]. Despite the global prevalence of GI parasites in wildlife [6], there has been limited research conducted on their prevalence in zoo animals within eastern China. Prevention and control of parasitic diseases in wildlife is the responsibility of zoo veterinarians [7]. To effectively evaluate and regulate the impact of intestinal parasites on animal populations, including zoonotic pathogens, it is essential to assess their prevalence within wildlife populations. This becomes even more crucial during the COVID-19 pandemic, as concerns regarding emerging zoonoses have heightened [8]. Furthermore, such assessments are vital for ensuring the health and safety of zoo veterinarians and tourists [9].

Traditionally, the detection and identification of Ascarididae eggs have relied on microscopy, which is a method that requires expertise and knowledge but can sometimes lead to identification errors [10,11]. For example, there is misidentification between *Toxocara canis* (Werner, 1782) eggs and *Toxocara cati* (Schrank, 1788) eggs, and some pollens are morphologically similar to Ascarididae eggs. However, in recent years, molecular techniques have emerged as valuable tools for distinguishing and classifying Ascarididae species [12]. Particularly, several genetic markers such as internal transcribed spacer 2 (ITS-2) in the nuclear ribosomal DNA (rDNA) region and cytochrome c oxidase subunit 1 (*cox*1) in the mitochondrial DNA (mtDNA) region have proven to be useful in investigating genetic diversity within the family Ascarididae [13,14,15]. However, Blouin [16] suggests that *cox*1 may provide more reliable results for genetic comparison compared to ITS in closely related nematode species.

In the present study, we aimed to determine the occurrence and variety of intestinal parasites in fecal samples collected from wild animals residing in the Zhuyuwan Zoo, located in the Yangzhou region of eastern China, and we utilized molecular genetic techniques to identify two Ascarididae species at the ITS-2 locus and subsequently compared their genetic divergence within an evolutionary tree at the *cox*1 locus. These results lay a foundation for prevention and control of wild animal parasitic diseases at a zoo in Jiangsu Province, eastern China.

## 2. Materials and Methods

### 2.1. Study Site

The study was conducted at Zhuyuwan Zoo in Yangzhou city, Jiangsu Province, located in eastern China (Figure 1). Yangzhou city, situated on the lower reaches of the Yangtze River, has a temperate continental monsoon climate with four distinct seasons. The area has a mild climate with ample sunshine and rainfall, which creates favorable conditions for the occurrence and transmission of parasitic diseases. The zoo relocated from Slender West Lake to Zhuyuwan in 2004. Currently, the zoo accommodates more than 40 species of mammals, birds, and amphibians from Asia, Africa, and the Americas.

### 2.2. Sample Collection

In 2021, a total of 104 fecal samples were randomly collected from different animals living in Zhuyuwan Zoo. Of them, 19 samples were collected from 15 species of birds, 19 from 8 species of primates, and 66 from 17 species of non-primate mammals. Trained animal handlers conducted the collection of fecal samples (~50 g) in the morning. The samples were stored in zipper bags and placed in a ~4 °C sample box for storage, and processed within 6 h of collection.

### 2.3. Microscopy

Nematode eggs and coccidian oocysts were detected using the saturated saline floatation method [17]. Briefly, 10 g of feces were diluted with 15 mL of saturated saline, filtered through a filter with a pore size of 250 μm, and the filtrate was centrifuged for 5 min at 800× *g*. A coverslip was placed over the surface of the supernatant, and it was viewed under a microscope after 3 min. The eggs of trematodes were examined using the nylon sieve washing method [17], i.e., 10 g feces diluted with water passed through 60 mesh (aperture = 250 μm) sieve and 260 mesh (aperture = 57 μm) sieve successively, and then the filter residue in the 260 mesh sieve was washed with water until the final filtrate was clear. Finally, the sediment in the sieve mesh was observed under a microscope. The eggs of nematode and trematode were observed and identified under a 40× objective lens. Unsporulated oocysts were incubated in 2.5% potassium dichromate (K_2_Cr_2_O_7_) for 5–7 days. After centrifugation at 800× *g* for 8 min, the supernatant was discarded, and the precipitate was resuspended in saturated saline and centrifuged at 700× *g* for 8 min. The sporulation oocysts from the supernatant were observed under the 40× objective. All identifications were performed as previously described [17,18,19,20]. Among nematode eggs, *Trichuris* and *Strongyloides* egg are easy to be identified according to their characteristic morphology and structure, in which *Trichuris* eggs are lemon shaped, yellow, or brown in color, with a thick smooth shell and a conspicuous polar plug at both ends, whereas *Strongyloides* egg are oval, thin-shelled and with larvae. In addition, the eggs of *Haemonchus* spp., *Ostertagia* spp., *Trichostronglus* spp. (belonging to Trichostrongylidae), and *Oesophagostomum* spp. (belonging to Cyathostomidae) possess similar sizes (73~95 μm × 34~50 μm), morphologies (ovoid in shape), and structures (containing numbers of embryo cells); it is difficult to distinguish between these species [17,18]. In this study, therefore, eggs with the above similar morphological structure were classified as Trichostrongylidae parasites.

### 2.4. DNA Isolation and PCR Amplification

About 100 eggs recovered from the African lions (*Panthera leo*) and the brown bears (*Ursus arctos*) were ground five times in liquid nitrogen, each time for 1 min. The subsequent steps for DNA extraction followed the guidelines outlined in the MiniBEST Universal Genomic DNA Extraction Kit Ver.5.0 (TaKaRa, Tokyo, Japan).

A standard PCR-based sequencing technique targeting ITS-2 locus (~300 bp, specific primers) was used to detect the *Toxascaris leonina* and *Baylisascaris transfuga* in the feces from an African lion and a brown bear, as previously reported [21,22]. Additionally, a standard PCR protocol was used to amplify the *cox*1 gene sequence, utilizing generic primers reported by Gasser et al. [19]. The expected size of the PCR product (*cox*1) was approximately 450 bp (Table 1). The PCR amplification process followed these cycling conditions: an initial denaturation step at 94 °C for 5 min, followed by 35 cycles of denaturation at 94 °C for 30 s, annealing at 60 °C (for ITS-2) or 50 °C (for *cox*1) for 30 s, extension at 72 °C for 60 s, and a final extension step at 72 °C for 5 min. Subsequently, we purified the PCR products using the MiniBEST DNA Fragment Purification Kit Ver.4.0 (TaKaRa, Tokyo, Japan) and subjected them to 1.0% agarose gel electrophoresis, followed by staining with ethidium bromide. The gel was then transilluminated and photographed using a gel imaging system (Bio-Rad, Hercules, CA, USA).

### 2.5. Sequence and Phylogenetic Analysis

All PCR products exhibited a single band. The purified PCR products were sent to Beijing Genomics Institute (BGI, Beijing, China) for Sanger sequencing. Subsequently, the sequencing results were analyzed using MegAlign with the clustal/W method and BLAST using highly similar sequences.

Phylogenetic analysis of *T. leonina* and *B. transfuga* was conducted based on the *cox*1 loci, and additional isolates from GenBank were included. The phylogenetic trees were constructed using the MEGA 5 software [23]. To determine the most suitable model, ModelTest in MEGA 5 was employed, and the Tamura-Nei [24,25] was utilized for Maximum Likelihood (ML), Neighbor-Joining (NJ), and Maximum Parsimony (MP) analyses. The reliability of the results was assessed through bootstrap analyses comprising 1000 replicates. *Trichuris suis* was selected as an outgroup.

## 3. Results

### 3.1. Occurrence of Intestinal Parasites

Based on egg morphology, a total of 11 species of parasites were identified, including *Fasciola* spp. in family Fasciolidae, *Paramphistomum* spp. in family Paramphistomatidae, *Capillaria* sp. in family Capillariidae, *Trichuris* spp. in family Trichuridae, *Strongyloides* spp. in family Strongyloididae, *Amidostomum* sp. in family Amidostomatidae, Trichostrongylidae, *T. leonina*, *B. transfuga* and *Parascaris equorum* (Goeze, 1782) in family Ascarididae, and *Eimeria* spp. in family Eimeriidae (Figure 2).

Of 11 species of parasites, Trichostrongylidae had the highest infection rate of 34.8% (23/104), followed by *Trichuris* spp. (5.8%, 6/104), *Eimeria* spp. (5.8%, 6/104), and *Strongyloides* spp. (3.8%, 4/104). The infection rate with other parasites was 0.9% (1/106). The positive rates for birds, primates, and non-primate mammals were 26.3% (5/19), 31.6% (6/19), and 50.0% (33/66), respectively (Table 2, Table 3 and Table 4). The occurrence of helminths and protozoans was 39.4% (41/104) and 5.8% (6/104), respectively (Table 5). Five fecal samples were mixed with two or more parasites with 5% (5/104) positivity rate.

### 3.2. PCR Amplification Analysis

A total of 23.9 ng/μL and 7.8 ng/μL of DNA were extracted from about 100 eggs collected from an African lion and a brown bear, respectively. At the ITS-2 locus, a band of 300 bp in size was amplified only using TleoF-NC2R specific primers from the eggs from an African lion (Figure 3A). Similarly, a band of 301 bp was amplified only using NITSF-NITSR primers from the eggs isolated from a brown bear (Figure 3B). The NC13-NC2 specific primers were employed as a control to verify the presence of parasite DNA in each sample. At the *cox*1 locus, a band of about 450 bp in size were successfully amplified using JB3-JB4.5 generic primers from *Toxascaris leonina* (African lion) and *Baylisascaris transfuga* (brown bear) and sequenced, respectively (Figure 3C). Afr L refers to samples recovered from African lion feces and Bn B refers to samples recovered from brown bear feces.

### 3.3. Phylogenetic Analysis

The homology analysis showed that the percentage of identity was 92.4% between *T. leonina* (MT895786) and *B. transfuga* (MT881703). *Toxascaris leonina* in this study (MT895786) shared 98.1% similarity with *T. leonina* from Changsha of China (MK522168, MK522175). The phylogenetic analysis revealed that *T. leonina* (MT895786) was grouped together with *T. leonina* (MT359318, MK522173, MK522175, MK522168 and JF780951) to form a sub-clade in the *cox*1 phylogenetic tree (Figure 4). *Baylisascaris transfuga* (MT881703) shared 100% similarity with *B. transfuga* (HM594948), and phylogenetic analysis revealed that *B. transfuga* (MT881703) was grouped together with *B. transfuga* (HM594948, HQ671079, EU628683, KY973960, KC543477, MF419818, EU628684) to form a sub-clade in the *cox*1 phylogenetic tree (Figure 4). Although the three algorithms (ML/NJ/MP) differed slightly in topology, all analyses yielded a consistent, robust phylogenetic resolution for *T. leonina* and *B. transfuga* and their congeneric species in the genera *Toxascaris* and *Baylisascaris*, repectively. The ML method was eventually adopted to construct this evolutionary tree. *Trichuris suis* (HQ183742) was used as an outgroup (Figure 4).

## 4. Discussion

The overall prevalence of GI parasites in animals living in Zhuyuwan Zoo was 42.3% (44/104), which is similar to the results observed in zoo animals in Xining of China [20], Slovenia (45%, 337/741) [21], and Poland (48%, 34/71) [22]. However, the infection rate of GI parasites in the present study was higher than that in twenty-four zoological gardens of China (26.51%, 317/1196) [23], Nepal (19.54%, 17/87) [24], and France (32.2%, 99/307) [25], and lower than that in Sichuan of China [26], Italy (80%, 24/30) [27], and Brazil (71.1%, 27/38) [28]. These differences may be related to the ecological environment and sampling season. The present study found a higher occurrence of helminths (39.4%, 41/104) compared to protozoans (5.8%, 6/104), similar to the observations in a zoological garden in Kenya, which revealed a higher occurrence of helminths (64.4%, 203/315) and a lower occurrence of protozoans (17.1%, 54/315) [29]. Interestingly, the prevalence of protozoa is lower than that of helminths in primates and non-primate mammals, whereas in birds the prevalence of protozoa infection is higher than that of helminths. This may be related to the environment in which these birds and primates and non-primate mammals live. These birds are wading birds whose feeding environment relies heavily on silt, which contains some fecal matter and some infectious *Eimeria* coccidia, thus making them susceptible to disease transmission [30]. Of course, it may also be due to birds being more susceptible to *Eimeria*, as *Eimeria* coccidia causes the most severe damage to poultry (belonging to birds). Additionally, our results revealed that the positive rates of GI parasites in birds, primates, and non-primate mammals were 26.3% (5/19), 31.6% (6/19), and 50.0% (33/66), respectively. Interestingly, the positive rate of mixed infection with two or more parasites was only 5% (5/104), which is consistent with previous reports (7%, 55/741) [21]. In addition, no tapeworms were detected in this study.

In this study, *Trichuris* spp. had a 5.8% (6/104) infection rate, second only to Trichostrongylidae (22.1%, 23/104). *Strongyloides* eggs were detected in fecal samples from the Ring-tailed lemurs (*Lemur catta*) and red pandas (*Ailurus fulgens*). *Strongyloides* parasites can infect their hosts through skin penetration or ingestion [31] and reproduce asexually in the host’s intestinal wall, contributing to their high infection rates [32]. Rondon, Ortiz, Leon, Galvis, Link and Gonzalez [33] reported that 21.6% (40/185) of fecal samples from neotropical primates were positive for *Strongyloides* spp. Studies on parasites infecting nonhuman primates are essential for better understanding the potential threat of zoonotic transmission to humans [34], especially given the ongoing processes of pervasive land use change and biodiversity loss [35] and the COVID-19 pandemic [36]. In addition, *Trichuris* spp., as a zoonotic parasite, may increase the risk of mixed infection with intestinal pathogens [37]. Therefore, Zhuyuwan Zoo needs to pay more attention to prevention and control of *Trichuris* spp. and *Strongyloides* spp.

The Père David’s deer (*Elaphurus davidianus*) and the sika deer (*Cervus nippon*) are National Protected Animals in China [38]. Here, *Paramphistomum* spp. eggs were first detected in the captive Père David’s deer in a zoo in China, and *Fasciola* spp. eggs were detected from the feces of sika deer. *Paramphistomum* spp. and *Fasciola* spp. have different degrees of pathogenicity to animals, and *Fasciola* parasites are an important zoonotic pathogen. So, Zhuyuwan Zoo needs to pay special attention to prevention and control of *Paramphistomum* spp. and *Fasciola* spp.

For decades, there has been considerable debate surrounding the systematics of members of the Ascaridomorpha (including Anisakidae, Ascarididae, Ascaridiidae and Toxocaridae) [39,40]. With the increasing utilization of molecular biology methods, there has been a growing ability to examine and identify ascarids [41]. This is particularly important because the eggs of many ascarid species share similar morphological characteristics, potentially leading to misidentification [42]. In the present study, two *Ascaris* species were identified using molecular biological methods in this study. It was found that *T. leonina* could only be amplified using *Tleo*F-NC2R primers and *B. transfuga* could only be amplified using NITSF-NITSR primers, respectively. The results of a phylogenetic tree showed that *T. leonina* was closer to *T. canis* than to *T. cati* at *cox*1 locus, which was similar to the result reported by Xie et al. [15].

*Toxascaris* spp. and *Baylisascais* spp., belonging to the family Ascarididae, are common GI parasites parasitized in carnivores such as members of Canidae, Felidae, and Ursidae. *T. leonina* mainly resides in the gastrointestinal tract of dogs, cats, and lions [43,44], which carries the potential risk to infect humans due to their close relationship with zoonotic ascarids like *Toxocara canis* and *Toxocara cati* [45,46]. *Baylisascaris transfuga* has been documented in all extant members of the family Ursidae worldwide, except for the spectacled bear [47,48]. The possibility of *B. transfuga* causing illness in humans has been suggested, and it poses a threat to public health [42,49]. Previous research has shown that *B. transfuga* larvae can migrate within the tissues of various animals like chickens [50], rabbits [51], mice [52], and *Mongolian jirds* [53] from their original locations in the intestine. However, the occurrence of visceral larva migrans (VLM) has only been observed in mammals [49]. Therefore, there is a potential risk for humans who come into contact with infectious eggs of *B. transfuga* in the environment [48]. In addition, the other five GI parasites detected in this study also are potentially dangerous for humans. A previous study has shown that anthelmintic drugs could effectively control soil-transmitted helminth infections [54]. Therefore, we recommend that zoos appropriately use albendazole or praziquantel for helminths eradication to reduce the risk of infection to tourists and veterinarians.

## 5. Conclusions

The prevalence of GI parasites was 42.3% (44/104) in Zhuyuwan Zoo in China. *Paramphistomum* spp. eggs were first detected from the captive Père David’s deer and *Fasciola* spp. eggs were first reported from sika deer in zoo in China. Ascaridoid parasite eggs from the African lions were identified as *T. leonina*, and those from the brown bear were identified as *B. transfuga*. Of 11 GI parasites detected in this study, seven are potentially dangerous for humans. The public health threat posed by these potential zoonotic parasitic agents requires attention.

## Figures and Tables

**Figure 1 animals-14-00375-f001:**
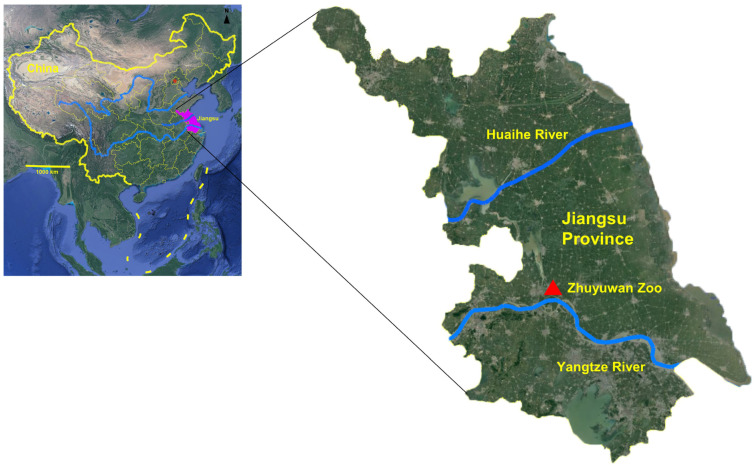
Geographical location of the sampling site, Zhuyuwan Zoo.

**Figure 2 animals-14-00375-f002:**
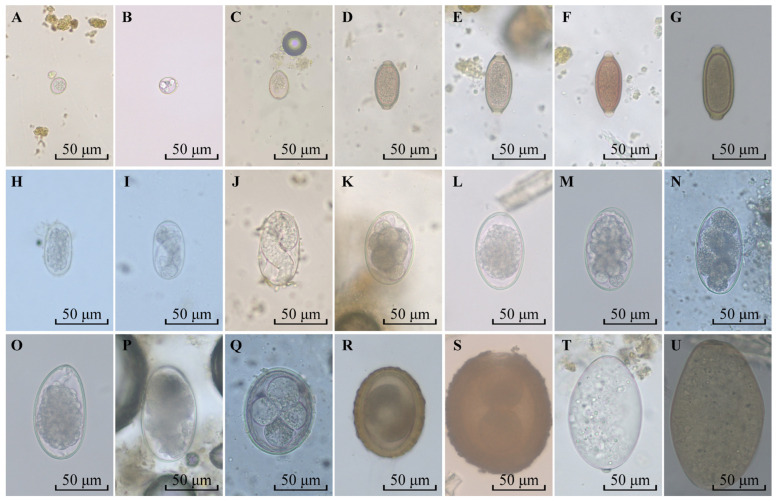
Parasites identified in stool samples from zoo animals. (**A**–**C**): *Eimeria* spp.; (**D**–**F**): *Trichuris* spp.; (**G**): *Capillaria* sp.; (**H**): *Amidostomum* sp.; (**I**,**J**): *Strongyloides* spp.; (**K**–**P**): Trichostrongylidae; (**Q**): *Toxascaris leonina*; (**R**): *Baylisascaris transfuga*; (**S**): *Parascaris equorum*; (**T**): *Paramphistomum* spp.; (**U**): *Fasciola* spp.

**Figure 3 animals-14-00375-f003:**
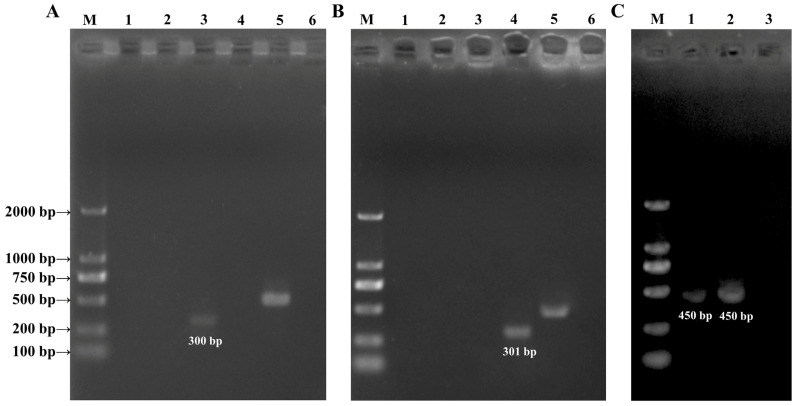
Results of amplification using PCR. (**Panel A**): amplification of SZ (300 bp) using ITS-2 primers; Lane 1: *Tocan*F-NC2R, Lane 2: *Tocat*F-NC2R, Lane 3: *Tleo*F-NC2R, Lane 4: NITSF-NITSR; Lane 5: NC13F-NC2R; Lane 6: negative control. (**Panel B**): amplification of ZX (301 bp) using ITS-2 primers; Lane 1: *Tocan*F-NC2R, Lane 2: *Tocat*F-NC2R, Lane 3: *Tleo*F-NC2R, Lane 4: NITSF-NITSR; Lane 5: NC13F-NC2R; Lane 6: negative control. (**Panel C**): amplification of SZ (450 bp) and ZX (450 bp) using *cox*1 primers; Lane 1: JB3-JB4.5; Lane 2 JB3-JB4.5; Lane 3: negative control.

**Figure 4 animals-14-00375-f004:**
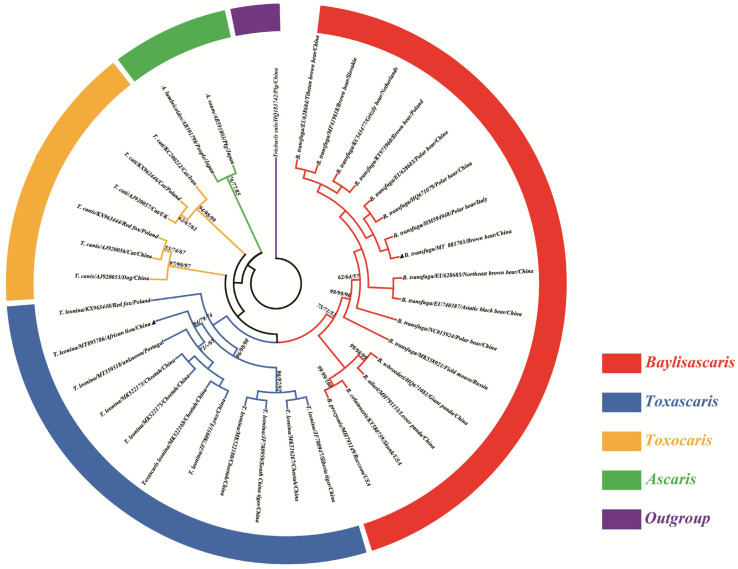
Evolutionary relationships of *T. leonina* and *B. transfuga* inferred using ML, NJ and MP analyses of *cox*1 sequences. *Toxascaris leonina* recovered from the African lion was genetically divided into the general *Toxascaris* and *B. transfuga* recovered from the brown bear was genetically divided into the general *Baylisascaris* which were indicated with blue and red rings, respectively. The general *Toxocaris* and *Ascaris* were indicated with yellow and green rings, and the Outgroup was purple, respectively. The numbers along branches indicate bootstrap values derived from different analyses in the order: ML/NJ/MP. Values lower than 50 are shown as “-”. Black triangles represent the sequencing results of this experiment.

**Table 1 animals-14-00375-t001:** Sequences of primers.

Name of Primer	Sequence (5′ to 3′)
For ITS-2	
TcanF	AGTATGATGGGCGCCAAT
NC2R	TTAGTTTCTTTTCCTCCGCT
TcatF	GGAGAAGTAAGATCGTGGCACGCGT
NC2R	TTAGTTTCTTTTCCTCCGCT
TaleoF	CGAACGCTCATATAACGGCATACTC
NC2R	TTAGTTTCTTTTCCTCCGCT
NITSF	TTATGAATTTTCAACATGGC
NITSR	GTTAGATGCTTAAATTCAGC
For *cox*1	
JB3	TTTTTTGGGCATCCTGAGGTTTAT
JB4.5	TAAAGAAAGAACATAATGAAAATG

**Table 2 animals-14-00375-t002:** Prevalence of intestinal parasites in birds at Zhuyuwan Zoo.

Species	*n*	No. (%) of Positive Samples for Parasite Species		
		*Capillaria* sp.	*Amidostomum* sp.	*Eimeria* spp.
Birds	19	1 (5.3)	1 (5.3)	3 (15.8)
Family Dromaiidae				
Emu (*Dromaius novaehollandiae*)	1	–	–	–
Family Struthioidae				
Common ostrich (*Struthio camelus*)	2	–	–	–
Family Phoenicopteridae				
Greater flamingo (*Phoenicopterus roseus*)	1	–	–	–
Family Gruidae				
Manchurian crane (*Grus japonensis*)	2	–	–	–
Hooded crane (*Grus monacha*)	1	–	–	1 (100)
White-naped crane (*Grus vipio*)	1	–	–	–
Siberian crane (*Grus leucogeranus*)	1	–	–	–
Black crowned-crane (*Balearica pavonina*)	2	1 (50)	–	–
Black-necked crane (*Grus nigricollis*)	1			1 (100)
Demoiselle crane (*Anthropoides virgo*)	1	–	–	–
Common crane (*Grus grus*)	1	–	–	–
Family Ciconiidae				
Oriental white stork (*Ciconia boyciana*)	1	–	1 (100)	–
Family Phasianidae				
Green peafowl (*Pavo muticus*)	1	–	–	1 (100)
Family Anatidae				
Northern pintail (*Anas acuta*)	2	–	–	–
Family Accipitridae				
Black vulture (*Aegypius monachus*)	1	–	–	–

*n* = number of samples collected and examined (the same below).

**Table 3 animals-14-00375-t003:** Prevalence of intestinal parasites in primates at Zhuyuwan Zoo.

Species	*n*	No. (%) of Positive Samples for Parasite Species	
		*Trichuris* spp.	*Strongyloides* spp.
Primates	19	3 (15.8)	3 (15.8)
Family Cebidae			
Black-capped capuchins (*Cebus apella*)	2	–	–
Squirrel monkey (*Saimiri sciureus*)	3	–	–
Family Lemuridae			
Ring-tailed lemur (*Lemur catta*)	4	–	3 (75)
Family Cercopithecidae			
François’s leaf monkey (*Trachypithecus francoisi*)	2	–	–
Golden monkey (*Rhinopithecus*)	2	2 (100)	–
Patas monkey (*Erythrocebus patas*)	1	1 (100)	–
Family Hylobatidae			
Lar gibbon (*Hylobates lar*)	3	–	–
Family Hominidae			
Chimpanzee (*Pan troglodytes*)	2	–	–

**Table 4 animals-14-00375-t004:** Prevalence of intestinal parasites in non-primate mammals at Zhuyuwan Zoo.

Species	*n*	No. (%) of Positive Samples for Parasite Species								
		Trichostrongylidae	*Trichuris* spp.	*Strongyloides* spp.	*Toxascaris leonina*	*Baylisascaris transfuga*	*Parascaris equorum*	*Paramphistomum* spp.	*Fasiola* spp.	*Eimeria* spp.
Non-primate mammals	66	23 (34.8)	3 (10.7)	1 (3.6)	1 (3.6)	1 (3.6)	1 (3.6)	1 (1.5)	1 (1.5)	3 (4.5)
Family Camelidae										
Llama (*Lama glama*)	2	1 (50)	–	–	–	–	–	–	–	–
Alpaca (*Vicugna pacos*)	1	1 (100)	1 (100)	–	–	–	–	–	–	–
Family Bovidae										
Blue *wildebeest* (*Connochaetes taurinus*)	11	4 (36.4)	–	–	–	–	–	–	–	–
Gemsbok (*Oryx gazella*)	10	10 (100)	2 (100)	–	–	–	–	–	–	2 (100)
Scimitar-horned oryx (*Oryx dammah*)	4	4 (100)	–	–	–	–	–	–	–	–
Family Equidae										
Pony (*Equus* spp., pony)	2	–	–	–	–	–		–	–	–
Common zebra (*Equus quagga*)	10	3 (30)	–	–	–	–	1 (10)	–	–	–
Family Cervidae										
Père David’s deer (Elaphurus davidianus)	1	–	–	–	–	–	–	1 (100)	–	–
Sika deer (*Cervus nippon*)	10	–	–	–	–	–	–	–	1 (10)	1 (10)
Family Ursidae										
Giant panda (*Ailuropoda melanoleuca*)	2	–	–	–	–	–	–	–	–	–
Brown bear (*Ursus arctos*)	2	–	–	–	–	1 (50)	–	–	–	–
Family Ailuridae										
Red panda (*Ailurus fulgens*)	2	–	–	1 (50)	–	–	–	–	–	–
Family Felidae										
Siberian tiger (*Panthera tigris* ssp. *altaica*)	2	–	–	–	–	–	–	–	–	–
African lion (*Panthera leo*)	2	–	–	–	1 (50)	–	–	–	–	–
Leopard (*Panthera pardus*)	1	–	–	–	–	–	–	–	–	–
Family Canidae										
Common wolf (*Canis lupus*)	2	–	–	–	–	–	–	–	–	–
Family Macropodidae										
Gray kangaroo (*Macropus giganteus*)	2	–	–	–	–	–	–	–	–	–

**Table 5 animals-14-00375-t005:** The overall occurrence of intestinal parasitic infections among various animals at Zhuyuwan Zoo.

Animals	Samples	Helminth Positive (%)	Protozoan Positive (%)	Total
Birds	19	2 (10.5)	3 (15.8)	5 (26.3)
Primates	19	6 (31.6)	0 (0)	6 (31.6)
Mammals	66	33 (50.0)	3 (4.5)	33 (50.0)
Total	104	41 (39.4)	6 (5.8)	44 (42.3)

## Data Availability

Data are contained within the article.

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
