# Peer review of "Prevalence of Gastrointestinal Parasites in Zoo Animals and Phylogenetic Characterization of *Toxascaris leonina* (Linstow, 1902) and *Baylisascaris transfuga* (Rudolphi, 1819) in Jiangsu Province, Eastern China"

_animals, 2024, doi:10.3390/ani14030375_

Round 1

Reviewer 1 Report

Comments and Suggestions for Authors

Dear authors,

Please find below some considerations for improving the manuscript.

Major

Tables 2-4 were not mentioned on the text.

The results described in the Results section are different from discussion:

Line 164: The occurrence of helminths and protozoans was 39.4% (41/104) and 5.8% (6/104), respectively.

Line 213: The present study found a higher occurrence of helminths (25.7%) compared to protozoans (7.6%).

Line 164 and Table 5: The occurrence of helminths and protozoans was 39.4% (41/104) and 5.8% (6/104), respectively. – However, in birds, Eimeria sp. is more frequent. I think that this should be discussed.

Minor

Line 30: Ascaris lumbricoides should be in italics.

Lines 190-197- Fig. 3 should be replaced by Fig. 4

Comments on the Quality of English Language

Only a minor review is necessary

Reviewer 2 Report

Comments and Suggestions for Authors

Major issues:

1. Section 2.2. A table should be provided, even if as supplementary information with the species sampled and with the parasites found for each species, as well as per group of species. Table 5 is mentioned in the Results (Section 3.1), but no table is included in the manuscript!

2. Section 2.5. "Phylogenetic analysis of T. leonina and B. transfuga was conducted" - why? Presumably it was of sequences obtained in this work, in relation to others. Explain in the manuscript why this analysis was necessary for these species.

3. In materials and methods, and then in Results, it should be stated if the primers are species generic or specific. 

4. Simple Summary, but also Summary: In "Phylogenetic analysis showed that Baylisascaris is more closely related to Toxacaris than to Ascaris.", is it Toxocara or Toxascaris? 

5. In any case, the phylogenetic tree produced does not support any relationships between the four different Genera, because the bootstrap support is low. And the authors do not mention these relationships in the Results section - only the Summary. So, the sentence is 262-264 should be removed from the Discussion and in 290-291 from the conclusions.

6. Figure 4 - What analysis was used to produce the tree shown? It cannot have been all methods, given that the authors mention some topological differences. If it is a consensus tree, then it should be stated. However, it would be better to show the tree produced by one method (ML, preferably) with branch lengths incorporated. 

7. Some results, such as mixed infections, are only mentioned in the Discussion.  The manuscript should be revised to include ALL results in the Results section.

Minor issues:

1. Line 68 - briefly indicate what kind of interpretation errors are more frequent. Misidentification between species, or with pollen, for example? Explain in the manuscript. 

2. Legend to Fig. 1 - It should show the location of the Zoo, as the animals were not collected from various locations in the wild. And it should be corrected to captive animals.

3. Section 2.5 - Indicate Sanger Sequencing, and what type of analysis was done with MegAlign and BLAST. Explain in the manuscript. 

4. Legend to fig. 2 and in the text - Trichostrongylidae is the name of a Family, so spp. is not used.

5. Lines 172 - how do you know here the species the eggs belong to, if there is no mention of morphology before, and before DNA sequencing? Explain in the manuscript. 

6. Legend to Fig. 3 - as there is no list of samples, and no explanation about the codes used for the samples, it is not clear what is present in each lane. However given the F and R notations, it seems like there are different primer pairs in each lane, which is rather odd. What is SZ and ZX? Explain in the manuscript. Also explain what was in the negative control, if different primers were used in the other reactions. 

7. lines 221-231: this paragraph makes more sense  in the methods section, in an abbreviated format.

8. Lines 232-235: this information is not needed in the Discussion. If, at all, it should be in Methods, to indicate how eggs were identified. 

9. Lines 232-248. Why are Strongyloides and Trichuris analyzed together in this paragraph? It is confusing and unnecessary.

Comments on the Quality of English Language

Correct the following: 

1. Toxacaris to Toxascaris, throughout the manuscript.

2. Toxocaris canis to Toxocara canis, throughout the manuscript.

3. At the beginning of a sentence, genus names are not abbreviated. So, for example, in line 272, B. transfuga (...) should be Baylisascaris transfuga (...)

4. Line 149 - "Based on egg morphology, a total of 11 types of parasites were identified..."

5. Write the family name after the word "family".

6. Lines 160-165  - review, in particular line 162, change to "The infection rate with other parasites..."

Reviewer 3 Report

Comments and Suggestions for Authors

Keeping animals in zoos increases the likelihood of parasitic infections, posing a significant threat to the animals themselves, as well as to people and local populations of wild and domestic animals. Therefore, this study on GI parasites in animals from Zhuyuwan Zoo (China) is always relevant. The manuscript undoubtedly makes a certain contribution to our knowledge of the GI parasites of zoo animals. The study of Chinese colleagues undoubtedly meets the goals and objectives of the journal Animals and can be published.

But I have some remarks about this manuscript, which will undoubtedly improve the article:

1. Line 17,25,33 - I would strongly recommend using the term ”species” here and in manuscript text. May be better to write: “More than 11 parasite species were identified.” Or rephrase it somehow differently.

Lines 29-30  Ascaris lumbricoides Linnaeus, 1758 is one species. It would be more correct to write here: “Two Ascaris species were molecularly characterized.” And don't forget about italics for Latin names of species and genera.

Line 39 – from the brown bear (Ursus arctos)

Line 68 – better use “identification errors”

Line 73 – The authors' initials are not included in the text – Blouin [16].

Figure 1 – This is a scientific article, not a political one. No need to show disputed territories on the map of China. In MDPI journals this may pass. However, you may have problems submitting subsequent articles to other journals. My advice is to enlarge the map of China or remove the borders.

The authors have only one sampling site - Zhuyuwan Zoo. Therefore the caption to Figure 1 needs to be changed. In this way: “Geographical location of the sampling site, Zhuyuwan Zoo.”

Lines 104,108 – format the reference correctly.

Lines 119 – Please, don't forget the article “the” – the African lions (Panthera leo), the brown bears (Ursus arctos). Please check all text. Further: The Ring-tailed lemurs (Lemur catta), the red pandas (Ailurus fulgens); the Père David's deer, etc.

Authors must submit the manuscript in accordance with the rules for formatting MDPI articles. Article title and the names of sections and subsections, all words in which must be capitalized (perhaps except for the parasite names). For example, “Prevalence of Gastrointestinal Parasites in Zoo Animals and Phylogenetic Characterization of Toxascaris leonina and Baylisascaris transfuga in Jiangsu Province, Eastern China” and “2.2. Sample Collection.”

According International Code of Zoological Nomenclature (ICZN) at the first mention of genera and species in the article text its full Latin name with the author and year of description should be given; in relation all species – parasites and their hosts (Toxascaris leonina (Linstow, 1902), Baylisascaris transfuga (Rudolphi, 1819), Panthera leo (Linnaeus, 1758); etc.). This should be done for all species in Latin in the article at the first mention. It is also appropriate to list in Table 1 the authors who described the species or genus.

Lines 150-154 – No need to write “family” every time. It is enough just to mention the family name in parenthesis: Fasciola spp. (Fasciolidae), Paramphistomum spp. (Paramphistomatidae), Capillaria sp. (Capillariidae)….

Line153 – Second mention of the species. Generic names must be abbreviated T. leonina, B. transfuga.

Caption to Figure 2. The authors studied GI parasites of zoo animals. What wild animals are we talking about? It would be better to name Figure 2: “Parasites identified in stool samples from zoo animals.” Or “captive animals”.

Line 147 – Correct the mistake.

Lines 161-165 – In section Materials and Methods you must indicate what infect index you used. Prevalence of infection?

Line 255 - Two Ascaris species …

Lines 259,260 – correct reference: Xie et al. [15] or Xie and coauthors [15]

Line 268 – … belonging to the family Ascarididae,

Line 272 - A sentence cannot begin with an abbreviated word. That's why here: ”Baylisascaris transfuga has been documented in…”

Authors’ contributions (lines 297) should be formalized according to MDPI journal rules.

The manuscript may be published in Animals, but some corrections are needed.

Comments on the Quality of English Language

English is understandable, however the manuscripts should be proofread by a professional translator or by a native speaker. A lot of sentences need be rephrasing. 

Round 2

Reviewer 1 Report

Comments and Suggestions for Authors

Dear authors,

The paper improved a lot. Congratulations.

Reviewer 2 Report

Comments and Suggestions for Authors

I am happy with the changes made by the authors.